# A simple clinical score to reduce unnecessary testing for Puumala hantavirus

Justus Brockmann[1,2], Michael Kleines[3], Narmin Ghaffari Laleh[4], Jakob Nikolas Kather[4], Stephanie Wied[5], Jürgen Floege[1], Gerald S. Braun[1,6,7]*

1 Department of Nephrology and Rheumatology, RWTH University Hospital Aachen, Aachen, Germany, 2 Department of Neurology, University of Leipzig, Leipzig, Germany, 3 Division of Virology, Center of Laboratory Diagnostics, RWTH University Hospital Aachen, Aachen, Germany, 4 Medical Faculty Carl Gustav Carus, Else Kroener Fresenius Center for Digital Health, TUD Dresden University of Technology, Dresden, Germany, 5 Department of Medical Statistics, RWTH Aachen University, Aachen, Germany, 6 Department of Nephrology, Klinikum Coburg, Coburg, Germany, 7 University of Split School of Medicine, Split, Croatia

* gebraun@ukaachen.de

## Abstract

### Background

Puumala hantavirus (PUUV) causes nephropathia epidemica (NE), an endemic form of transient acute renal injury (AKI). Serological testing is the mainstay of diagnosis. It was the aim of the present study to assist decision-making for serological testing by constructing a simple tool that predicts the likelihood of PUUV positivity.

### Methods

We conducted a comparative cohort study of all PUUV-tested cases at Aachen University tertiary care center in Germany between mid-2013 and mid-2021. N = 293 qualified for inclusion; N = 30 had a positive test result and clinical NE; N = 263 were negative. Two predictive point scores, the Aachen PUUV Score (APS) 1 and 2, respectively, were derived with the aid of logistic regression and receiver operating characteristic (ROC) analysis by determining the presence of four admission parameters. For internal validation, the internal *Monte Carlo* method was applied. In addition, partial external validation was performed using an independent historic cohort of N = 41 positive cases of NE.

### Results

APS1 is recommended for clinical use as it estimated the probability of PUUV positivity in the entire medical population tested. With a range from 0 to 6 points, it yielded an area under the curve of 0.94 by allotting 2 points each for fever or headache and 1 point each for AKI or LDH>300 U/L. A point sum of 0–2 safely predicted negativity for PUUV, as was confirmed in the NE validation cohort.

**Data Availability Statement:** The complete minimal datasets of both the primary and the validation patient cohort and the code for the score generation and additional information on

parameters of ROC analysis are all posted on: https://github.com/narminGhaffari/Hanta-Virus-Project The complete minimal data set tables will also be sent with the revision for the reviewers to be viewed directly.

**Funding:** The author(s) received no specific funding for this work.

**Competing interests:** The authors have declared that no competing interests exist.

## Conclusion

Here, we present a novel, easy-to-use tool to guide the diagnostic management of suspected PUUV infection/NE and to safely avoid unnecessary serological testing, as indicated by point sum class 0–2. Since 67% of the cohort fell into this stratum, half of the testing should be avoidable in the future.

## Introduction

*Hantaviridae* are single-stranded, enveloped RNA viruses of the order *Bunyavirales*, which are transmitted by rodents all over the world [1,2]. They cause several distinct clinical presentations in humans worldwide [3–7]. In Northwestern Europe, the subtype Puumala (PUUV) prevails, causing >10,000 annual cases of nephropathia epidemica (NE), manifesting as a milder form of hemorrhagic fever with renal syndrome (HFRS), almost invariably transmitted by the bank vole [8–10]. Fever, headache, lumbar pain, acute kidney injury (AKI) and thrombocytopenia comprise well-established hallmarks of the disease [5,11,12]. AKI is usually transient with a favorable ten-year outcome that is comparable to that of never-infected individuals [13]. The incidence of NE is seasonal and oscillates in multi-annual cycles coupled to the size of the zoonotic vector population [14]. The latter is in turn determined by climatic and nutritive conditions. In Germany, the number of reported outbreaks of PUUV infections and NE has increased during the last two decades, with around 10,000 cases being reported over this period [15,16]. This development appears both reflective and causative of an increased awareness towards NE and possibly due to climatic changes as well. Here, we explored whether there was an over-consideration and over-testing for PUUV in the hospital workup of patients presenting with AKI [17].

The main purpose of this study was to i) estimate the adequacy of serologic testing for hantavirus in the case of a German tertiary care hospital and to ii) derive a tool for the differential diagnostic workup of NE based on simple routine clinical parameters. To this end, we first established our study population's baseline characteristics and, as a second step, we built on that information for score construction.

## Material and methods

### Study population

Approval, including the waiver of the need for informed consent, was obtained from the institutional review board (IRB) of the University of Aachen Medical Faculty (EK 416–20 on November 11, 2020) and from the University Hospital's data safety officer (CTC-A Nr. 20–285 on December 4, 2020). The primary study population was identified by searching the electronic data base of the hospital's department of virology (*Molis* software, CGM, Koblenz, Germany) to identify the complete set of hospitalized cases aged ≥18 years subjected to serologic testing for hantavirus between mid-2013 and mid-2021. Consecutively, the corresponding administrative and clinical data were extracted from the hospital's electronic patient record data base (*Medico* software, Oracle-Cerner, Berlin, Germany). This data retrieval was performed from December 7, 2020 to June 30, 2021 by a team consisting of MK, GSB and JB, who initially had access to information that could identify individual participants. For subsequent analyses and for the remaining authors, anonymization was ensured by use of a lookup table. A total of N = 293 (N = 30 clinically confirmed PUUV-positive and N = 263 PUUV-negative)

cases remained for final analysis after excluding incomplete data sets (**Fig 1**). For partial validation, a historic independent unpublished cohort of N = 41 confirmed NE cases with full available status information on fever, headache, AKI and LDH was used. The cohort was extracted from the total N = 57 positively tested cases collected in our center from 2001–2012, the analysis of which was also covered by the same IRB approval. N = 16 cases had to be excluded for incompleteness of score input data. The comprehensive primary data sheets of both cohorts

**Fig 1. Study flow diagram summarizing patient identification, exclusion and subsequent analysis.** Absolute number of patients as indicated.

have been posted in Microsoft Excel format on: https://github.com/narminGhaffari/Hanta-Virus-Project.

Located in northwestern Germany, RWTH Aachen University hospital treats an average of 45,000 hospitalized patients per year at a tertiary care level. PUUV infection/NE is endemic in this region at a low-to-intermediate rate with an incidence of 0,5-10/100,000 patients per year [14].

## Laboratory testing

For serologic testing of the main cohort (2013–2021), the *Recomline Hantaplus IgG and IgM* [R] kit (Mikrogen, Germany) was applied, which allows for the detection and differentiation of immunoglobulin (Ig)G and IgM antibodies specific for the hantavirus variants PUUV, SNV (Sin Nombre), HTNV (Hantaan), DOBV (Dobrava) and SEOV (Seoul). Testing of the validation cohort (2001–2012) was performed at the time by immunofluorescence with subsequent Western blot for PUUV, HTNV and SEOV. In all hanta-positive cases of both cohorts, PUUV was the identified variant. Urinary sediment for hematuria and standard urine dipstick test results for hematuria and proteinuria were converted into semiquantitative ranks as follows: (-) 0, (+/-) 1, (+) 2, (++) 3, (+++) 4, (++++) 5, (+++++) 6. A value $\geq 1$ was defined as hematuria and proteinuria, respectively.

## Definition of acute kidney injury (AKI)

The presence of AKI was defined in an adaption of the Kidney Disease: improving Global Outcomes (KDIGO) definition [18] as an $\geq$0.3 mg/dL elevation of the admission serum creatinine compared to the laboratory's upper limit of normal (ULN; 1.0 mg/dL in females and 1.2 mg/dL in males). This definition was chosen since the scores developed herein aim at an acute setting where preclinical creatinine values are often unavailable. To verify the accuracy of this approach, we used a double retrospective validation method. We calculated i) AKI on admission according to an $\geq$0.3 mg/dL or $\geq$1,5-fold elevation using the lowest available creatinine during hospital stay and ii) development of AKI at any stage by comparing the highest and the lowest available creatinine during hospital stay. As a result, only 5% of cases were not supported by either one of the retrospective methods, while 40% were supported by both and 55% by at least one method. This level of inaccuracy was deemed acceptable. The corresponding calculations can be found in the comprehensive primary data sheet in cells X299:AG307, as posted on: https://github.com/narminGhaffari/Hanta-Virus-Project.

## Basic statistical analysis

Data are presented as absolute numbers, medians with interquartile range or percentages. For the comparison of binomial variables between the PUUV-positive and -negative groups, Fisher's exact test was used in the case of few counts $\leq$5 (Tables 1 and 2 and Fig 3), and the Chi-square test was applied in the case of frequent counts >5 (Table 3). For the comparison of quantitative discrete or continuous variables, the two-tailed t-test with Welch's correction for unequal variances was used (Tables 1 and 2 and Fig 2). For the comparison of categorical variables, the two-tailed Mann-Whitney U test was employed (Table 1). To compare the respective minimum platelet counts or the maximum serum creatinine levels of hantavirus patients with those of hantavirus-negative disease groups, analysis of variance (ANOVA) was applied (Fig 5). A p-value of $\leq$0.05 was considered as statistically significant. Analyses were computed in Prism9 (GraphPad, San Diego, USA).

**Table 1. Study Population and admission parameters.**

| | PUUV positive N = 30 | PUUV negative N = 263 | N available for analysis per group | P-value |
|---|---|---|---|---|
| Age [years], *median (IQR)* | 46 (31–54) | 54 (34–66) | 30, 263 | **0.03*** |
| Gender [male patients], *N* | 22 (73%) | 168 (64%) | 30, 263 | 0.30 |
| Blood pressure [mmHg], *median (IQR)* | 123/76 (107/72-132/84) | 125/75 (113/67-141/83) | 23, 168 | 0.22 |
| WBC [/nL; 4.0–10.0], *median (IQR)* | 10.0 (8.4–11.9) | 9.2 (6.9–13.1) | 30, 263 | 0.81 |
| Platelets [/nL; 150–400], *median (IQR)* | 89 (74–249) | 197 (118–271) | 30, 263 | 0.09 |
| Hb females [g/dL; 12–16], *median (IQR)* | 13.6 (12.0–14.3) | 11.4 (9.4–12.8) | 8, 95 | **0.01*** |
| Hb males [g/dL; 14–18], *median (IQR)* | 13.8 (12.4–15.5) | 12.3 (10.1–13.9) | 22, 168 | **0.008**** |
| LDH [U/L; <250], *median (IQR)* | 325 (300–392) | 268 (205–390) | 28, 260 (26, 233)¶ | 0.14 (**0.001****)¶ |
| PCT [mg/dL; 0.5–1.4], *median (IQR)* | 1.6 (0.7–2.9) | 0.6 (0.2–3.4) | 16, 150 (14, 117)¶ | 0.86 (**0.02***)¶ |
| CRP [mg/L; <0.5], *median (IQR)* | 57 (39–90) | 38 (10–110) | 24, 238 (23, 206)¶ | 0.63 (0.07)¶ |
| Creatinine [mg/dL; 0.5–1.2], *median (IQR)* | 2.1 (1.6–5.7) | 2.1 (1.0–4.3) | 30, 263 (30, 232)¶ | 0.40 (**0.02***)¶ |
| Proteinuria [rank 0–6], *median (IQR)* | 3 (0–5) | 2 (0–3) | 27, 203 | **0.05** |
| Hematuria [rank 0–6], *median (IQR)* | 3 (2–4) | 3 (2–5) | 27, 204 | 0.37 |

Abbreviations: [], unit and reference range; CRP, C-reactive protein; Hb, hemoglobin; *IQR*, interquartile range; LDH, lactate dehydrogenase; *N*, absolute number of patients; PCT, procalcitonin; PUUV, acute Puumala hantavirus infection status; WBC, white blood count.

Statistical testing: All quantitative discrete and continuous variables by two-tailed t-test with Welch's correction; ordinal variables (i.e., *hematuria* and *proteinuria*) by two-tailed Mann-Whitney U-test; binomial variables (i.e., *gender*) by Fisher's exact test

*p≤0.05

**p≤0.01.

¶ alternative analysis following removal of outliers using Prism9 software, see Methods.

## Score construction

Harnessing various combinations of the clinical and laboratory discriminating parameters, as depicted in **Fig 3**, a multivariable logistic regression model was applied to predict PUUV-positive serology in patients. The rounded β-regression coefficients were used to calculate the risk factors for each patient and were employed as multiplication factors (weight) in the construction of our score. Akin to a previous study by Latus et al., where the number of risk factors met

**Table 2. Study population outcome parameters.**

| | PUUV positive N = 30 | PUUV negative N = 263 | N available for analysis per group | P-value |
|---|---|---|---|---|
| Intensive care required, *N* | 1 (3%) | 65 (24%) | 30, 263 | **0.005**** |
| Ventilatory support required, *N* | 0 (0%) | 34 (12%) | 30, 263 | **0.03*** |
| Renal replacement therapy required, *N* | 3 (10%) | 53 (20%) | 30, 263 | 0.23 |
| Death, *N* | 0 (0%) | 17 (7%) | 30, 263 | 0.23 |
| Platelet minimum [/nL; 150–400], *median (IQR)* | 89 (74–248) | 149 (85–222) | 30, 263 | 0.88 |
| Creatinine maximum [mg/dL; 0.5–1.2], *median (IQR)* | 4.8 (2.0–6.7) | 2.4 (1.1–5.1) | 30, 263 | **0.04*** |

Abbreviations: [], unit and reference range; IQR, interquartile range; N, absolute number of patients; PUUV, acute Puumala hantavirus infection status.

Statistical testing: All binomial variables by Fisher's exact test; quantitative continuous variables (i.e., *platelet minimum* and *creatinine maximum*) by two-tailed t-test with Welch's correction

*p≤0.05

**p≤0.01.

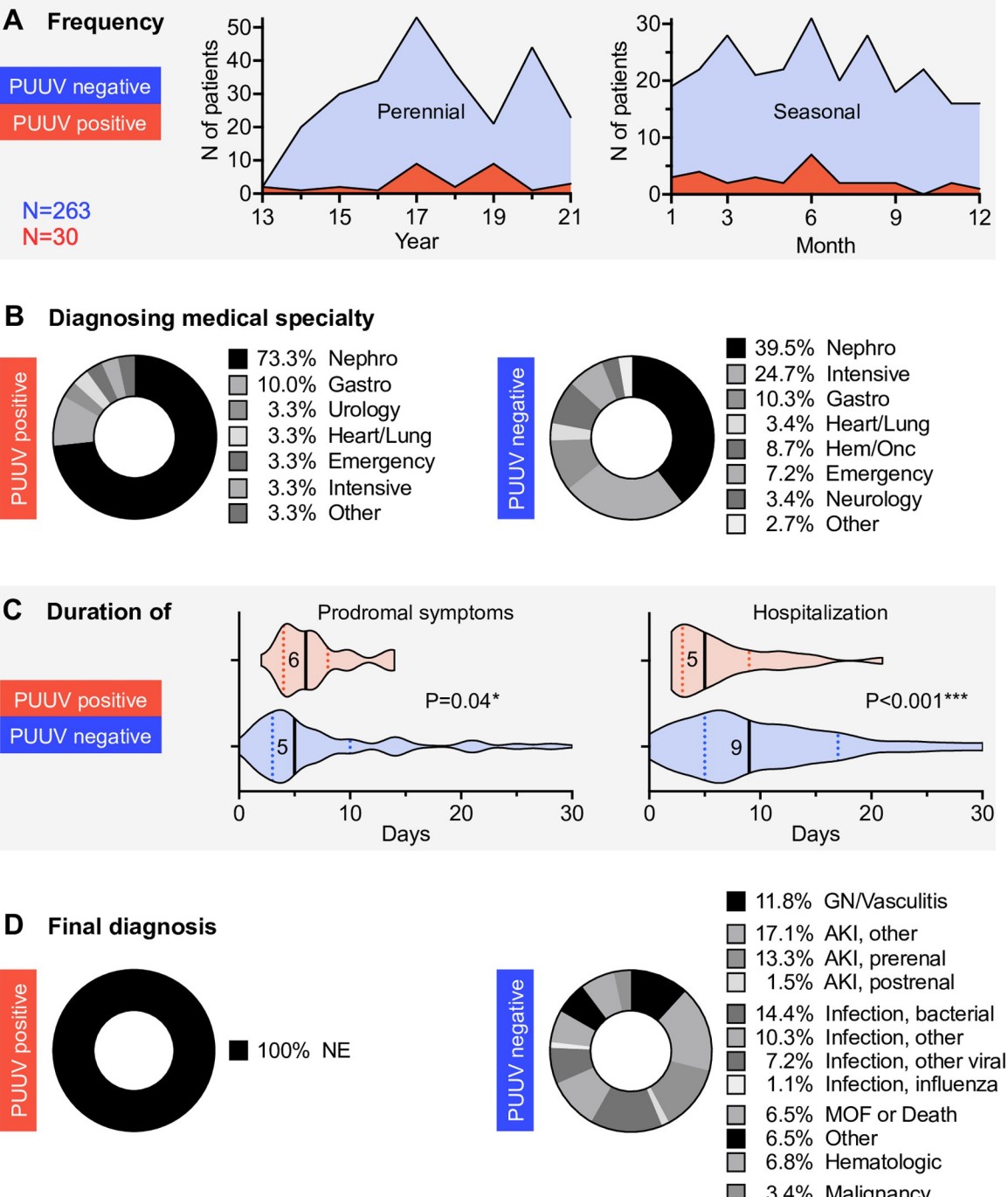

**Fig 2. Patient demographics stratified by acute Puumala hantavirus infection (PUUV) status.** Positives indicated in red, negatives in blue. (A) Perennial and seasonal cumulative case incidences of years mid-2013 to mid-2021. (B) Medical specialties involved in the initial diagnostic management and instigation of serological testing. (C). Durations of prodromal symptoms and lengths of stay. Data are medians with interquartile ranges. (D) Compilation of final diagnoses. *Abbreviations*: AKI, acute kidney injury as defined by KDIGO/ Kidney Disease: Improving Global Outcomes; AKI, other, AKI of neither GN/Vasculitic, nor pre-, or postrenal etiology; Emergency, emergency department; Gastro, gastroenterology; GN/Vasculitis, glomerulonephritis and vasculitis; Heart/Lung, cardiology and pulmonary medicine; Hem/Onc, hematology and oncology; Intensive, intensive care unit; Infection, other, neither bacterial nor viral cause identified; Infection, other viral, viral infection not caused by influenza; MOF, multiple organ failure; N, absolute number of patients available for analysis; NE, nephropathia epidemica; Nephro, nephrology; PUUV, acute Puumala hantavirus infection status. *Statistics*: Analyzed group sizes were N = 30 and N = 263 for positives and negatives, respectively, except N = 25 and N = 127, respectively, for prodromal length. Two-tailed t-test with Welch's correction; *p≤0.05, ***≤0.001.

## A  Admission symptoms

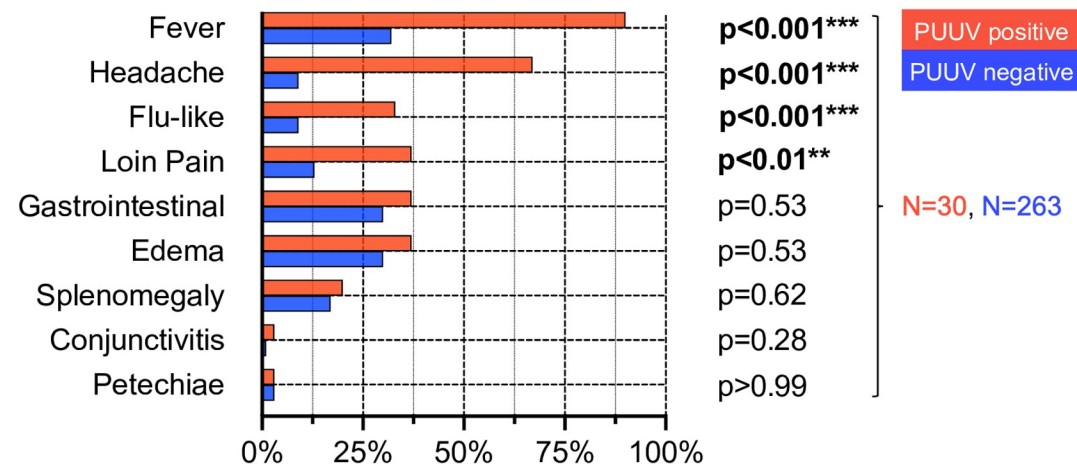

## B  Admission laboratory abnormalities

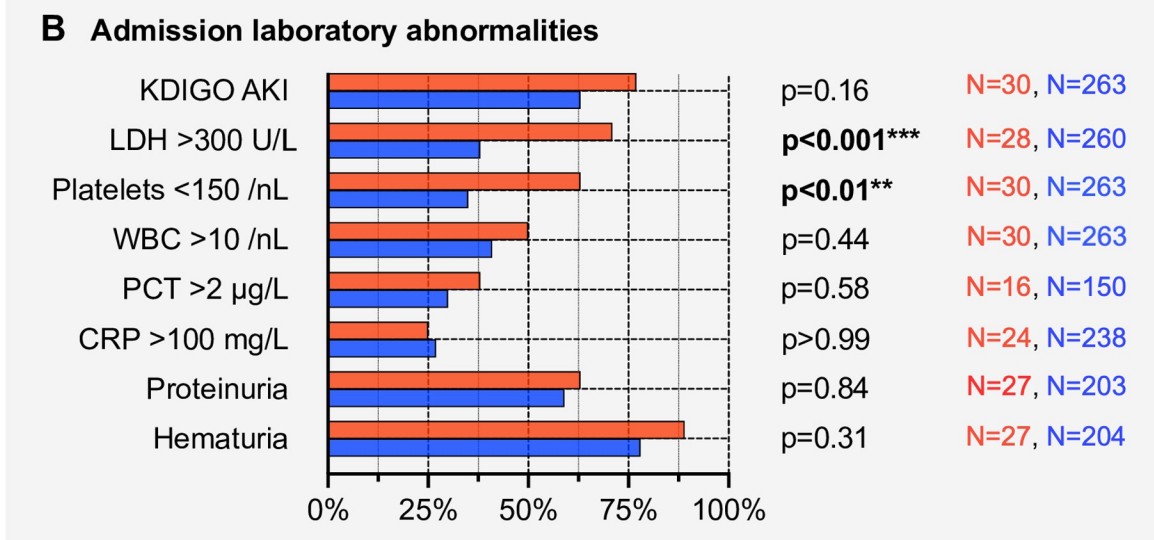

**Fig 3.** Statistical evaluation of symptoms (A) and laboratory abnormalities (B) on admission–hantavirus positives and negatives indicated by red and blue linework, respectively. *Abbreviations*: CRP, C-reactive protein; KDIGO AKI, acute kidney injury as defined by Kidney Disease: Improving Global Outcomes; LDH, lactate dehydrogenase; N, absolute number of patients available for analysis; PCT, procalcitonin; PUUV, acute Puumala hantavirus infection status; WBC, white blood count. *Statistical testing*: Fisher's exact test; **p≤0.01, ***≤0.001.

was used to estimate the AKI risk with the aid of *receiver operating characteristic (ROC)* analysis, here the number of risk factors multiplied by their respective weight was used to estimate the probability of a positive hantavirus serology [19]. As a result of our screening process, the four parameters, fever, headache, AKI and serum lactate dehydrogenase (LDH) level >300 U/L, were selected to derive two risk-index scores that could predict the probability of PUUV infection, i.e., the Aachen PUUV Score (APS) 1 and 2. The rationale for choosing these four parameters lies in their respective highest statistical performance and in the consideration that AKI was to be included as the primordial hallmark of acute hantavirus infection. The difference between the APS 1 and 2 scores is based on the control group used for generation: either the entirety of PUVV negatives or the PUUV negatives with proven non-hanta viral infection,

**Table 3. Contingency tables of diagnostic hit rate and PUUV score class by medical specialty.**

| | Nephrology | Other specialties | Total, N | P-value |
|---|---|---|---|---|
| **Diagnostic hit rate of the serologic tests initiated by respective specialty** | | | | |
| PUUV positive, N | 22 (17.5%) | 8 (4.8%) | 30 | <**0.001**\*\*\* |
| PUUV negative, N | 104 (82.5%) | 159 (95.2%) | 263 | |
| Total, N | 126 (100.0%) | 167 (100.0%) | 293 | |
| **Score 1 point sum classes of the individuals tested by respective specialty** | | | | |
| 0–1 points, N | 60 (47.6%) | 61 (36.5%) | 121 | 0.056 |
| 2–6 points, N | 66 (52.4%) | 106 (63.5%) | 172 | |
| Total, N | 126 (100.0%) | 167 (100.0%) | 293 | |

Abbreviations: PUUV, acute Puumala hantavirus infection status; N, absolute number of patients.

Statistical testing: Chi-square test

\*\*\*p≤0.001.

respectively. This resulted in different weights being allocated to the clinical criteria. **S1 Table** shows the β-regression coefficients of a variety of models.

All datasets were complete, except for LDH, where N = 3 missing values were imputed using the 'most frequent' method, which in this case meant that it was posited to be not elevated. The cutoff for LDH was identified by a subgroup analysis of mean, median and percentiles. Since it exhibited the best performance among other classification algorithms such as *Support Vector Machine* (SVM) [20], *Decision Tree* [21] and *KNeighborhood* [22], logistic regression [23] was used for further evaluation. The fitted models provided a competent classification with an area under the curve, as shown in the graphs. ROC analysis and score development was implemented in Python 3. The detailed code has been posted on: https://github.com/narminGhaffari/Hanta-Virus-Project.

## Validation of the ROC models

Considering the limited size of our data set, computing a single *area under the curve (AUC)* would not have allowed for sufficient confidence in the performance of the models. Unfortunately, a second full data set as a validation cohort was not available to us. Thus, as a first step, the *Monte Carlo* cross-validation method was used [24]: a training set was created by randomly selecting 80% of the data without replacement, while the remaining 20% of the data formed the test set. One primary and an additional N = 5 iterative runs were performed using a different seed for randomization each time, to generate the 'single-fold' and 'five-fold'/'mean' ROC curves, respectively. The procedure is graphically summarized in **S1 Fig**. Unlike *k-fold cross-validation*, *Monte Carlo* allows for the potential overlap between iterations. Detailed insights into this methodology, including the rationale and execution, are also available within the code itself.

## Partial validation of the score's negative-prediction performance

While a second independent data set of positive and negative patients systematically tested for PUUV by serology was unavailable to us, we had at our disposal an unpublished historic collective of N = 41 acutely PUUV-positive only individuals with corresponding clinical and laboratory parameters from the years 2001–2012 (see above under 'Study population'). Using this cohort, APS1 and APS2 were calculated to define and verify the negative prediction of PUUV status.

## Results

### Baseline characteristics

Over the course of 8 years of hantavirus testing in adults at our institution, a total of N = 30 individuals were found to be positive and confirmed to have acute-type PUUV infection with clinical signs of NE, while the diagnosis was excluded in N = 263 of the cases (**Table 1**). This corresponded to an approximate ratio of 1:9. The median age in both groups was around 50±4 years. On average, PUUV-positive patients presented with low platelets, elevated LDH, and elevated procalcitonin (PCT).

The clinical course of NE patients was associated with a higher intensity and rate of AKI compared to PUUV-negative cases, as indicated by the maximum serum creatinine levels attained while, on the other hand, a substantial proportion of PUUV-negative cases required organ support or died (**Table 2**). Severe AKI, when present, resolved more rapidly in NE cases (not shown).

The occurrence of NE/PUUV-positive cases oscillated perennially with a seasonal focus in the first half of the year, which was in line with the literature [14] and most likely due to the mild, humid climate during the winter season in this region of Germany (**Fig 2A**). Positive cases were diagnosed predominantly by the Department of Nephrology (**Fig 2B**). While prodrome lengths until medical hospitalization were significantly longer for PUUV positives, PUUV negatives exhibited a significantly longer average length of stay (by 80%) (**Fig 2C**). Final diagnoses in the negative group were roughly equally distributed between AKI, infection, and miscellaneous causes (**Fig 2D**).

### Two Aachen PUUV Scores (APS)

Following the above assessments, relevant clinical and laboratory admission parameters were organized in a binomial manner (**Fig 3**). The most powerful discriminators between positivity and negativity for PUUV were i) fever, ii) headache, iii) elevated LDH >300 U/L and iv) thrombocytopenia <150/nL in that order. Given both its prevalent role in the presentation of PUUV infection and its strength as a coefficient in score construction, AKI was also considered for scoring. In a series of explorative modellings, each of these five criteria, with the exception of thrombocytopenia, added in a meaningful way to score performance. The **S1 Table** shows the β-regression coefficients of explorative scorings providing an overview of their strength, choice and the applied rounding. In the end, two Aachen PUUV prediction scores (APS) were then chosen for in-depth analysis (**Fig 4**). The disutility of thrombocytopenia is probably best explained by its narrow window of detection requiring an early stage of medical presentation with NE leading to only intermediate frequency. This is underscored by the fact that the thrombocyte mean nadir was not significantly different between PUUV-positive and -negative subjects studied (**Table 2**).

APS1 (**Fig 4A–4D**) compares the PUUV-positive/NE group with the entire PUUV-negative group consisting of a broad variety of diagnoses associated with instigation of hantavirus serological testing in our hospital. According to its β-coefficients, APS1 allocates 2 points each to the clinical criteria of fever and headache, and 1 point each to the laboratory criteria of AKI or LDH >300 U/L on admission, respectively, yielding a total maximum of up to 6 points (**Fig 4A**). The corresponding operating characteristic curve (ROC) and test validity data suggest a solid score performance from point sum class 3 onwards (**Fig 4B**). For clinical use, a given patient's point sum class is simply computed and the corresponding pretest probability ranges are derived (**Fig 4C**). An APS1 point sum class within the range of 0 to 2 is associated with a very low, 3 (0–7) %, maximum probability of a positive hantavirus infection. Since 67%

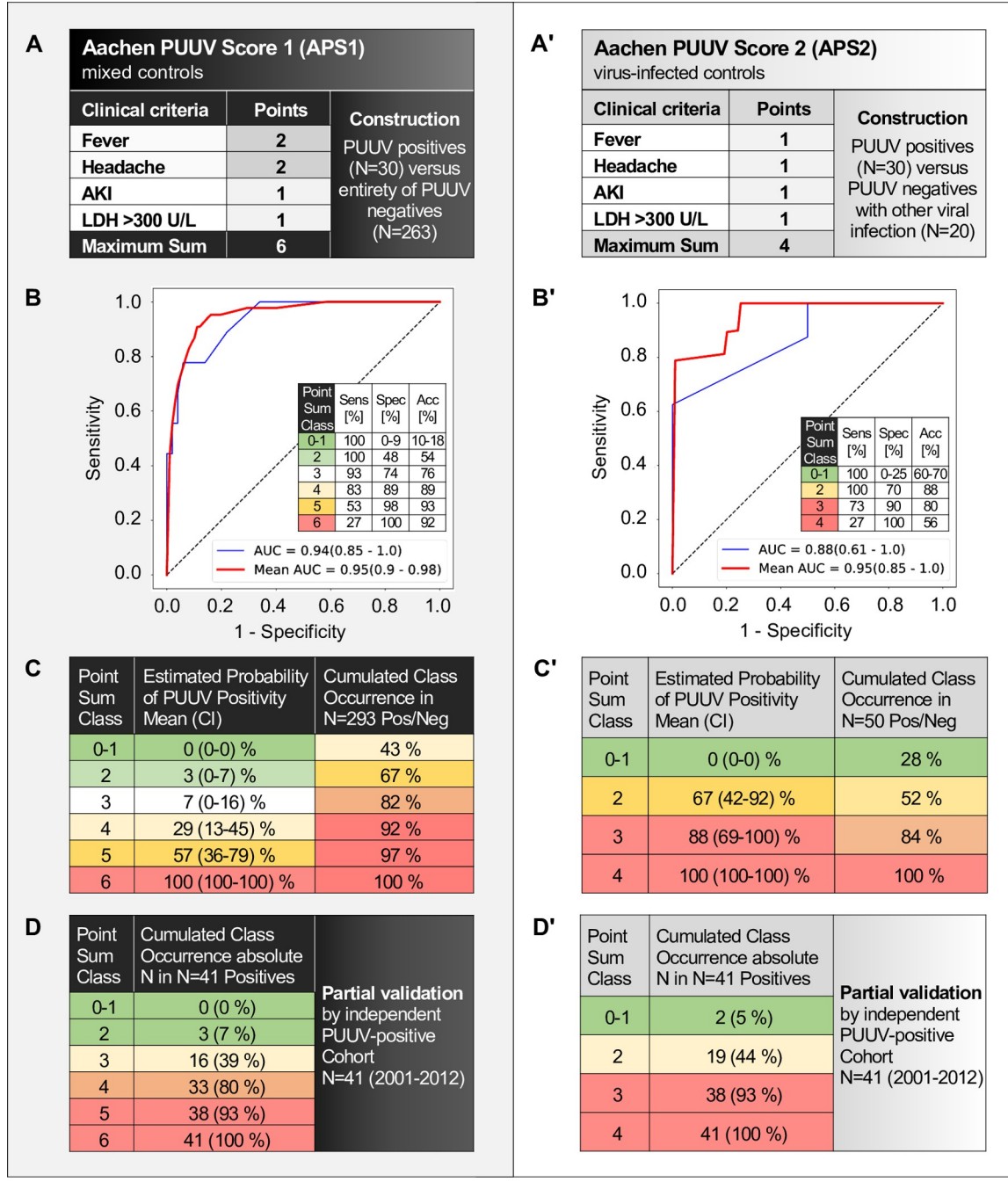

**Fig 4. Construction and partial validation of two alternative scores for the prediction of PUUV positivity.** (A-C) Aachen PUUV Score (APS) 1, derived using the complete control group of heterogeneous mixed diagnoses. (A'-C') APS2, derived using a control group consisting of only confirmed other viral infections. (A, A') Basic information and scoring tables on the assignment of score points per criterium. (B, B') Receiver operating and the corresponding test characteristics of sensitivity, specificity, and accuracy. (C, C') Results tables displaying for each point sum class its i) corresponding estimated probability of PUUV positivity and the actual PUUV positivity ii) the occurrence of point sum classes in a cumulated fashion in the complete cohort. (D, D') Partial score validation of APS1 and APS2. Use of an independent historic cohort of N = 41 PUUV-positive-only subjects with acute hantavirus infection from the years 2001–2012. Population characteristics as indicated in S2 Table. Shown are occurrences of point sum classes. Importantly, point sum class 0–1 predicted (C, C') and confirmed (D, D') a 0–5% probability of PUUV positivity in both scores, as highlighted in green shade, thus defining it as a rule-out criterium. In APS1, point sum class 2 was also associated with a very low mean probability of PUUV positivity of 3% and thus also highlighted in green shade. Highlighted in red shade are point sum classes associated with predicting or confirming a >90% probability of positivity. *Abbreviations*: Acc, accuracy; AKI, acute kidney injury according to KIDIGO/Kidney Disease: Improving Global Outcomes; AUC, area under the curve; LDH, lactate dehydrogenase; N, absolute number of patients per group; PUUV, acute Puumala hantavirus infection status; in brackets: confidence intervals. Sens, sensitivity; Spec, specificity.

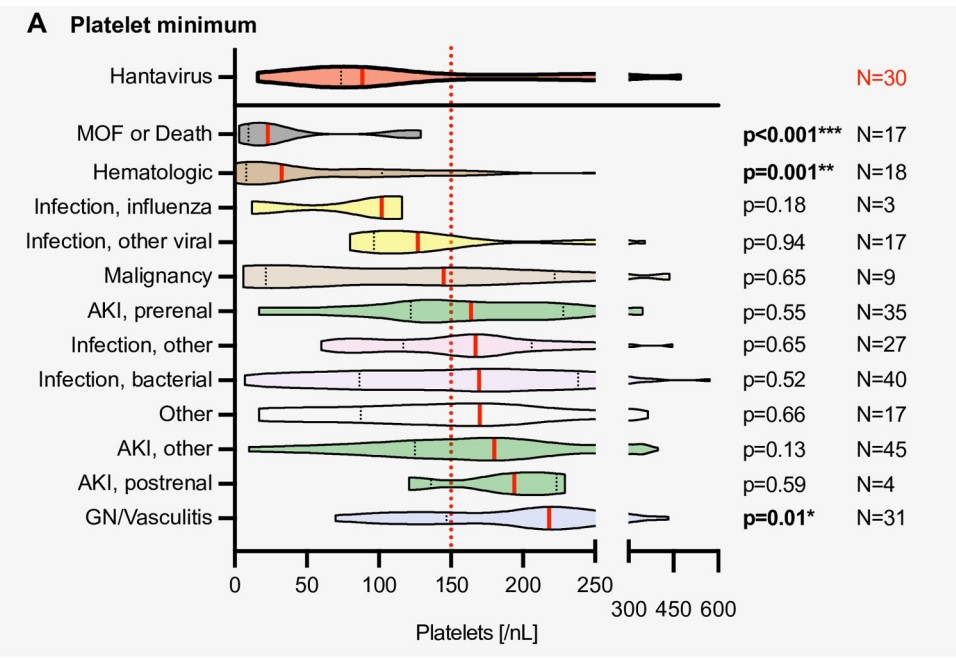

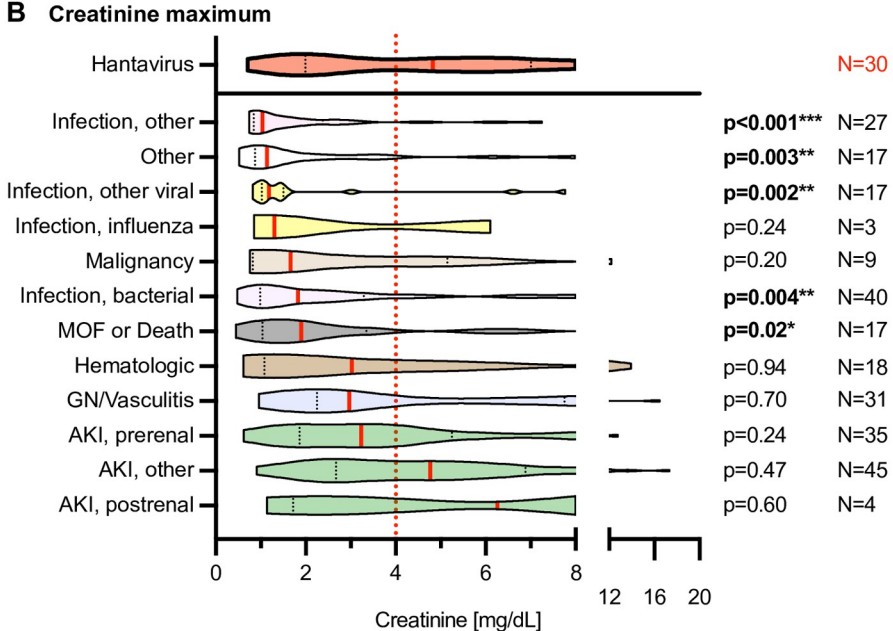

**Fig 5.** Diagnosis-specific comparison of the key follow-up parameters platelet minimum (A) and creatinine maximum (B). Hantavirus group is shown on top, highlighted in red. *Abbreviations*: AKI, acute kidney injury as defined by KDIGO/Kidney Disease: Improving Global Outcomes; AKI other, AKI of neither GN/Vasculitic, nor prerenal, or postrenal etiology; GN/Vasculitis, glomerulonephritis and vasculitis; Infection, other, neither bacterial nor viral cause identified; Infection, other, viral, viral infection not caused by influenza; MOF, multiple organ failure; N, absolute number of patients per group. *Statistical testing*: ANOVA; *p≤0.05, **≤0.01, ***≤0.001.

of patients fell into this group, there appears to be a substantial amount of unnecessary testing that could be potentially avoided in the future by application of APS1 scoring (**Fig 4C**). In conjunction with the test validity data, this further corroborates the usefulness of a threshold of 0–2 points to discount, and a cutoff of 3 or more points to consider, the possibility of

hantavirus infection, respectively. In addition to the intrinsic *Monte Carlo* validation method, we performed a validation of the PUUV-positive score arms using an independent historic cohort of N = 41 confirmed NE cases with status records on fever, headache, AKI and LDH treated at our center between 2001–2012. The cumulated point sum class occurrence in this cohort (**Fig 4D**) was strikingly similar to the probabilities of positivity associated with point sum classes of APS1 (**Fig 4C**). Further details on the characteristics of this historic PUVV-positive-only partial validation cohort are provided in **S2 Table**. In our view, APS1 is adequate for everyday clinical use when a swift decision must be made while a deeper differential diagnostic workup is not yet available. The variety in this control group is further showcased by its restriction to an infection-related etiology in only half of the cases and its diverse course of laboratory parameters over time (**Fig 5**).

APS2 (**Fig 4A'–4D'**) was derived by comparing the PUUV-positive group with the control subset of confirmed viral infections. The rationale for its development was to explore if there are differences when hantavirus infections are directly compared to other viral infections, a comparison that appears very logical but is less frequently encountered in the real world. According to its derived β-coefficients, APS2 simply allocates 1 point each per criterion (**Fig 4A'**). As can be also seen in **S1 Table**, this indicates that fever and headache become less important discriminators in this comparison, while AKI becomes slightly more important in terms of β-coefficients. ROC and score validity data (**Fig 4B'**), score data (**Fig 4C'**) and corroboration data from the second cohort (**Fig 4D'**) are analogous to APS1. Similar to APS1, point sum class 0–1 is associated with zero and 5% probability of PUUV positivity in APS2 and the partial validation cohort, respectively. This corresponds to 28% of tested cases, again suggesting that at least one third of the testing is avoidable.

## Analysis of serologic testing behavior

Nephrology was the medical specialty primarily detecting PUUV-positive/NE patients. We wondered if there might be significant differences in the pick-up rate or average serologic testing threshold among nephrology and other medical specialties. To address this question, we generated two contingency tables separating the diagnostic hit rates and PUUV score classes from these two groups of medical specialties (**Table 3**). Indeed, compared to other specialists, nephrologists exhibited a significant, about 3.5 times higher, success rate in the detection of PUUV-positive cases among instigated serologies (4.8% vs 17.5%, p<0.001). However, at the level of potentially avoidable testing, kidney specialists performed no better than their counterparts. The proportion of patients with 0% probability of positivity (as ascertained by a PUUV score classification of 0–1) was 47.6% in the hands of a nephrologist and 36.5% in those of other medical specialists based on the total of cases these disciplines tested, i.e., N = 126 and N = 167, respectively (p = 0.056).

## Discussion

The differential diagnostic awareness for hantavirus has substantially increased during the last two decades, owing to the characterization of endemic regions, increasing prevalence of PUUV infections, publication of seminal analyses and increasing availability of diagnostic tests [15,16]. While early recognition of NE helps to reduce unnecessary invasive procedures, which may amount to up to 25% [17], this development, from a theoretical standpoint, may also drive serologic over-testing leading to additional costs and decisional delays.

We developed the Aachen PUUV Scores (APS) starting from the longest period of comprehensive positive and negative PUUV serologic records available to us. Both of the generated APS methods require only two clinical (fever and headache) and two laboratory (acute kidney

injury/AKI and elevated lactate-dehydrogenase/LDH >300 U/L) admission criteria to be computed (**Fig 4A and 4A'**). Prediction studies by others have addressed a comparison with leptospirosis [25–28] or used the methodology of matched controls [29]. Importantly, the latter study also identified fever, headache and elevated LDH as the most significant laboratory findings associated with positivity for hantavirus, thus corroborating our chosen parameters.

As the central takeaway of our study, we have shown that point sum class 0–1 or even 0–2 can effectively rule out NE using our scoring systems (**Fig 4C and 4C'**), which was also validated in an independent cohort (**Fig 4D and 4D'**). APS1, which we recommend for application in clinical practice as outlined in the results section, harbors the best cutoff points: 0–2 as an indication for exclusion and above 3 points indicating a PUUV infection should be considered. As showcased in the 67% of tested individuals in the former group with very low probability of PUUV positivity, there is a substantial potential for the future avoidance of unnecessary tests by application of APS1. According to our analyses of other medical specialist subgroups that instigated diagnosis, the potential for avoiding unnecessary serology is universal to all disciplines (**Table 3**).

Our rationale to develop two alternative scores was driven by our acknowledgement that inter-physician and inter-center variances in test decisions directly influence the shape of a score. At one end of the variance spectrum, we developed APS1 with a control group featuring a low pretest probability of NE, given that much of its testing was prompted by a propensity for differential diagnostic completeness rather than convincing cues for NE. Unsurprisingly, in APS1, fever and headache as signs of infection were allotted a higher relative weight of 2 versus 1 point by the ROC analysis. To generate an alternative perspective featuring a higher pretest probability of NE, we chose the alternative criterium of a severe systemic viral illness and constructed APS2 by restricting the control group to cases of proven viral infection. Indeed, the allotted weights of fever and headache as discriminators between PUUV positivity and negativity were no longer higher, but on par with AKI and LDH >300 U/L. In summary, however, we recommend the use of APS1 in the majority of clinical settings. We consider APS2 to be most effective in giving an important additional perspective when corroborating the overall validity of APS1.

Our study's limitations lie in its sample size despite a long collection period, its retrospective design and the fact that we were unable to find a full external cohort for further validation. Consequently, score performance data to rule in PUUV (rather than out) remain limited. The reliability of our study's data recovery, however, appears completely on par with literature data on sex, clinical severity, fever [5,29,30], splenomegaly, procalcitonin [31,32], platelet count time courses [19], headache (our study 67% versus 68%) and elevated LDH (our study 71% versus 57–55%) [17,33]. The high predictive strength of these latter two parameters may indicate that mild encephalitis and loss of cell membrane integrity are common in PUUV infection [34,35].

## Conclusion

We present two overlapping Aachen PUUV scores for the prediction of the probability of Puumala hantavirus infection in a general and a more viral differential diagnostic workup setting, respectively. We propose our scores be chiefly applied by forgoing serological testing with a score point sum class of 0–1, or more widely with an APS1 point sum class of 0–2. Application of this criterion should result in a reduction of serologic testing by at least ∼50%, based on the characteristics of the cohort analyzed herein.

## Supporting information

**S1 Fig. Schematic of the *Monte Carlo* method applied.**
(TIF)

**S1 Table. Comparison of potential scores with corresponding β-coefficients.**
(PDF)

**S2 Table. PUUV-positive-only validation cohort of N = 41 (2001–2012).**
(PDF)

**S1 Dataset.**
(XLSX)

**S2 Dataset.**
(XLSX)

## Author Contributions

**Conceptualization:** Gerald S. Braun.

**Data curation:** Justus Brockmann.

**Formal analysis:** Justus Brockmann, Narmin Ghaffari Laleh, Jakob Nikolas Kather.

**Investigation:** Justus Brockmann, Michael Kleines.

**Methodology:** Justus Brockmann, Narmin Ghaffari Laleh, Stephanie Wied, Gerald S. Braun.

**Project administration:** Gerald S. Braun.

**Resources:** Jürgen Floege.

**Software:** Jakob Nikolas Kather.

**Supervision:** Gerald S. Braun.

**Validation:** Justus Brockmann, Gerald S. Braun.

**Visualization:** Justus Brockmann.

**Writing – original draft:** Justus Brockmann.

**Writing – review & editing:** Justus Brockmann, Jürgen Floege, Gerald S. Braun.

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
