## [Decision Letter · Decision Letter 0]

27 Nov 2023

PONE-D-23-22222A simple clinical score to reduce unnecessary testing for Puumala hantavirusPLOS ONE

Dear Dr. Braun,

Thank you for submitting your manuscript to PLOS ONE. After careful consideration, we feel that it has merit but does not fully meet PLOS ONE’s publication criteria as it currently stands. Therefore, we invite you to submit a revised version of the manuscript that addresses the points raised during the review process.

We appreciate the interesting and well conducted study. However, there are some minor points raised by the reviewer#2’s. Please carefully respond to the reviewer comments and suggestions.

We look forward to receiving your revised manuscript.

Kind regards,

Vipa Thanachartwet, M.D.

Academic Editor

PLOS ONE

3. Please upload a copy of Supporting Information Figure (S1 Fig.) which you refer to in your text on page 6.

Additional Editor Comments:

Additional comments to the authors are shown as follows:

For statistical analysis, the comparison of quantitative discrete or continuous variables, the two-tailed t-test with Welch's correction for unequal variances was used (Table 1, Table 2, and Figure 2). This test should meet the assumption of equal means, unequal variances and follow normal distribution, but the data show as median (IQR). Please carefully check. Missing values were imputed using the most frequent method (Page 7). What kind of most frequent method use for missing values, i.e., this may be mean, median or mode to represent the average value of the population for missing value. Please clearly define that would be clear for the readers.Please check Figure 2D. The combination of PUUV negative was 97%, this should be 100%.

Reviewers' comments:

Reviewer's Responses to Questions

**Comments to the Author**

1. Is the manuscript technically sound, and do the data support the conclusions?

Reviewer #1: Yes

Reviewer #2: Partly

2. Has the statistical analysis been performed appropriately and rigorously? 

Reviewer #1: Yes

Reviewer #2: Yes

3. Have the authors made all data underlying the findings in their manuscript fully available?

Reviewer #1: Yes

Reviewer #2: Yes

4. Is the manuscript presented in an intelligible fashion and written in standard English?

Reviewer #1: Yes

Reviewer #2: Yes

5. Review Comments to the Author

Reviewer #1: I appreciate being chosen to review the research article on Puumala Hantavirus, titled "A Simple Clinical Score to Reduce Unnecessary Testing," written by Brockmann J, Kleines M, Ghaffari Laleh N, Kather JN, Wied S, Floege J, and Braun GS.

A new tool created by authors predicts the likelihood of PUUV infection, aiding clinicians in deciding on serological testing for hantaviruses.

The manuscript is a comparative cohort study and it is generally well written. The study is well-methodologically organized and is written in standard English. The aim is clearly defined.

The study's results provided gainful insights into the adequacy of serologic testing for hantavirus in the case of a German tertiary care hospital. They derived a tool for the differential diagnostic workup of nephropathia epidemica (NE) based on simple clinical routine parameters. Data analysis and statistics are performed to a high technical standard and are described in sufficient detail. The conclusions are presented in an appropriate fashion and are supported by the research results. The study provides important clinical results that can guide the diagnosis of suspected PUUV infection/NE and help avoid unnecessary serological testing.

The references referred to by the authors are current.

The research meets all applicable standards for the ethics of experimentation and research integrity.

The authors gave relevant responses to the questions/doubts of both previous reviewers.

I am pleased to recommend that the manuscript "A simple clinical score to reduce unnecessary testing for Puumala hantavirus," be accepted for publication without further corrections.

Reviewer #2: PLOS ONE

November 6, 2023

Manuscript ID: PONE-D-23-22222

Manuscript title: A simple clinical score to reduce unnecessary testing for Puumala hantavirus

Dear Prof. Vipa Thanachartwet,

This study was conducted as a comparative cohort study of all PUUV-tested cases in Germany between 2013 and mid-2021 with the aim to assist decision-making for serological testing by constructing a simple Aachen PUUV Score (APS) that predicts the likelihood of PUUV positivity. The two predictive point scores were constructed. APS 1 estimated the probability of PUUV positivity in the entire medical population tested and APS 2 narrowed the estimation down for subjects with a confirmed viral infection. The present study constructed a novel, easy-to-use tool to guide the diagnostic management of suspected PUUV infection/NE and to safely avoid unnecessary serological testing.

However, there are comments for this manuscript to address as follows:

1. As fever, headache, lumbar pain, acute kidney injury (AKI) and thrombocytopenia comprise well-established hallmarks of the disease (Page 3), the result showed that the most powerful discriminators between positivity and negativity for PUUV were i) fever, ii) headache, iii) elevated LDH >300 U/L and iv) thrombocytopenia <150/nL in that order (Page 10). It is wonder why thrombocytopenia did not include for scoring. In addition, AKI was not significant different between the 2 groups. Why was it included in scoring? The explanation for these two variables are unclear.

2. APS 1 compares the PUUV-positive/NE group with the entire PUUV-negative group consisting of heterogeneous diagnoses associated with instigation of hantavirus serological testing in our hospital. APS 2 compares the former with the latter’s subset of confirmed viral infections. In both scores, it is unclear how to weight for scores (Figure 4). The statistical analysis for this part are not clearly defined in the results.

3. Point sums of at least 3 and 6 were associated with a >90 probability of PUUV positivity, in Score 1 and Score 2, respectively (Page 12). This statement is unclear because the maximum of score 1 is 6 and the maximum of score 2 is 4.

4. The intrinsic Monte Carlo validation method was performed to validate the PUUV-positive score arms using an independent historic cohort of N=41 confirmed NE cases with status records on fever, headache, AKI and LDH treated our center between 2001-2012. No data were available to support the statements. How many simulations were performed and how many rounds of each simulation?

5. Figure 5 did not showed in the result part, but this figure showed in discussion.

6. Please check for miss-spelling throughout the manuscript.

7. In page 14, we developed the Aachen PUUV Scores (APS) starting from the longest period of comprehensive positive and negative PUUV serologic records available to us. Both the generated APS require only two clinical (fever and headache) and two laboratory (acute kidney injury and elevated lactate-dehydrogenase >300 U/L) admission criteria to be computed (Fig 4B, B’). The abbreviation should be used. Please check.

These are all issues raised for this manuscript that would be clear for the readers and minor revision is needed.

6. PLOS authors have the option to publish the peer review history of their article (what does this mean?). If published, this will include your full peer review and any attached files.

Reviewer #1: No

Reviewer #2: **Yes: **Assoc. Prof. Varunee Desakorn

---

## [Author Response · Author response to Decision Letter 0]

30 Apr 2024

Brockmann et al., authors’ point-by-point responses to reviewer #2’s comments

Please find our answers point-by-point below the respective referee points (sometimes abbreviated):

REFEREE QUESTION 1. As fever, headache, lumbar pain, acute kidney injury (AKI) and thrombocytopenia comprise well-established hallmarks of the disease (Page 3), the result showed that the most powerful discriminators between positivity and negativity for PUUV were i) fever, ii) headache, iii) elevated LDH >300 U/L and iv) thrombocytopenia <150/nL in that order (Page 10). It is wonder why thrombocytopenia did not include for scoring. In addition, AKI was not significant different between the 2 groups. Why was it included in scoring? The explanation for these two variables are unclear.

ANSWER: We thank the reviewer for this logical question. The score criterion of AKI on admission has been included very deliberately as a result of the first peer review of 2021. The underlying concept is that a score for the in- or exclusion of a condition should also comprise the most common presentation of this condition, which in this case is AKI. In addition, the inclusion of AKI performed well in our scores.

To clarify this, we have now expanded explanatory text in the methods section (entire pages 7,8) and in the results section (pages 11 and 13). We also provide novel S1 Table, stating the beta-coefficients of various comparative score analyses. The test runs included both thrombocytopenia as a fifth parameter in addition to AKI and thrombocytopenia as a fourth parameter instead of AKI. The bottom line of these analyses was, as can be derived from the table, that the chosen four parameters presented herein provided the best trade-off of robustness and easy usability. The complete minimal data set allowing for reproduction of the figures of the N=293 cohort is now also posted on the github-platform: https://github.com/narminGhaffari/Hanta-Virus-Project

The following reflections may illustrate this further: 

i) the absence on average of a significantly different admission AKI-rate among the PUUV-positive and -negative groups underscores the high frequency of this key criterion in both groups. This also means that the “control group” of PUUV-negatives was adequate. AKI then performed better than expected in the beta-coefficient analysis because of the case-specific distribution of AKI in combination with the other criteria. 

ii) thrombocytopenia on the other hand performed less favorable. This is likely due to the fact that, while it features a practical 100%-prevalence in the course of NE, it can often no longer be detected upon presentation of an individual patient to the hospital due to its brief window of detectability (of only a couple of days early in the disease). 

REFEREE QUESTION 2. In both scores, it is unclear how to weight for scores (Figure 4). The statistical analysis for this part are not clearly defined in the results.

ANSWER: We thank the reviewer for this remark. See our answers to 1 this respect. In addition, we have rearranged Figure 4 to increase its readability at first glance. 

REFEREE QUESTION 3. Point sums of at least 3 and 6 were associated with a >90 probability of PUUV positivity, in Score 1 and Score 2, respectively (Page 12). This statement is unclear because the maximum of score 1 is 6 and the maximum of score 2 is 4.

ANSWER: We are now treating APS1 and APS2 completely separately in the results section, thus avoiding confusion on points.

REFEREE QUESTION 4. The intrinsic Monte Carlo validation method was performed to validate the PUUV-positive score arms using an independent historic cohort of N=41 confirmed NE cases with status records on fever, headache, AKI and LDH treated our center between 2001-2012. No data were available to support the statements. How many simulations were performed and how many rounds of each simulation?

ANSWER: Please note that we performed Monte Carlo intrinsically on the main cohort of N=293 subjects, and then, independently, as a second step validation by mere calculation using the N=41 positive-only validation cohort. To make this more transparent to the reader, we have now included: 

i) S1 Figure. It represents a graphical summary of the Monte Carlo method and how it was applied herein. We are very grateful to the reviewer for this remark, since we believe that it has prompted us to provide a much clearer presentation of the paper, also in our own view. 

ii) S2 Table. It provides the imminent details on the historic N=41 PUUV-positive only cohort from 2001-2012. The minimal data set of this cohort is also posted on github: https://github.com/narminGhaffari/Hanta-Virus-Project

REFEREE QUESTION 5. Figure 5 did not showed in the result part, but this figure showed in discussion.

ANSWER: The introduction of Fig 5 has now been shifted to the results section.

REFEREE QUESTION 6. Please check for miss-spelling throughout the manuscript.

ANSWER: Thank you, this has now been done by use of a professional English native-speaking corrector. 

Among other corrections, a mis-reference of Fig 5 as a former S1 Fig has been corrected. 

In addition, we have carefully re-checked our data. In the course of this action, in the interest of more clarity and robustness, we have also redefined AKI in a more prospective and simple way. As a result, we have recalculated all data and scores.

The exact tracked changes can be found in the manuscript version with highlighted changes compared to the previous version. The main outcome and message of the paper has not been altered by this. The figures and conclusions are similar. As stated above, the complete minmal data set files of both the main 2013-2021 and the 2001-2012 verification cohorts have been made available on github: https://github.com/narminGhaffari/Hanta-Virus-Project

REFEREE QUESTION 7. In page 14, we developed the Aachen PUUV Scores (APS) starting from the longest period of comprehensive positive and negative PUUV serologic records available to us. Both the generated APS require only two clinical (fever and headache) and two laboratory (acute kidney injury and elevated lactate-dehydrogenase >300 U/L) admission criteria to be computed (Fig 4B, B’). The abbreviation should be used. Please check.

ANSWER: Thank you, this has now been done.

FINAL SUMMARY BY REFEREE / EDITOR: These are all issues raised for this manuscript that would be clear for the readers and minor revision is needed.

ANSWER: We would like to thank the reviewer for her thorough and useful comments. In summary, this has led us to re-address the paper with respect to its clarity and readability. It has also led us to the inclusion of additional data (S1 Table, S1 Figure, S2 Table, complete minimal datasets for both cohorts) and to re-calculate the entire analyses using a more robust AKI identification formula (as stated in the methods on pages 5-6).

---

## [Decision Letter · Decision Letter 1]

14 May 2024

A simple clinical score to reduce unnecessary testing for Puumala hantavirus

PONE-D-23-22222R1

Dear Dr. Braun,

We’re pleased to inform you that your manuscript has been judged scientifically suitable for publication and will be formally accepted for publication once it meets all outstanding technical requirements.

Kind regards,

Vipa Thanachartwet, M.D.

Academic Editor

PLOS ONE

Additional Editor Comments (optional):

The authors response all issues raised by the reviewers.

Reviewers' comments:

Reviewer's Responses to Questions

**Comments to the Author**

1. If the authors have adequately addressed your comments raised in a previous round of review and you feel that this manuscript is now acceptable for publication, you may indicate that here to bypass the “Comments to the Author” section, enter your conflict of interest statement in the “Confidential to Editor” section, and submit your "Accept" recommendation.

Reviewer #2: All comments have been addressed

2. Is the manuscript technically sound, and do the data support the conclusions?

Reviewer #2: Yes

3. Has the statistical analysis been performed appropriately and rigorously? 

Reviewer #2: Yes

4. Have the authors made all data underlying the findings in their manuscript fully available?

Reviewer #2: Yes

5. Is the manuscript presented in an intelligible fashion and written in standard English?

Reviewer #2: Yes

6. Review Comments to the Author

Reviewer #2: All issues have been addressed. The manuscript is much improved and it is easy to understand. The results would help clinicians in decision making to do the serological test.

7. PLOS authors have the option to publish the peer review history of their article (what does this mean?). If published, this will include your full peer review and any attached files.

Reviewer #2: **Yes: **Assoc. Prof. Varunee Desakorn

---

## [Editor Report · Acceptance letter]

21 May 2024

PONE-D-23-22222R1 

PLOS ONE

Dear Dr. Braun, 

I'm pleased to inform you that your manuscript has been deemed suitable for publication in PLOS ONE. Congratulations! Your manuscript is now being handed over to our production team.

Kind regards, 

on behalf of

Professor Vipa Thanachartwet 

Academic Editor

PLOS ONE